# Comparison on the Aging of Woods Exposed to Natural Sunlight and Artificial Xenon Light

**DOI:** 10.3390/polym11040709

**Published:** 2019-04-18

**Authors:** Ru Liu, Hanwen Zhu, Kang Li, Zhong Yang

**Affiliations:** 1Research Institute of New Forestry Technology, Chinese Academy of Forestry, Haidian, Beijing 100091, China; liuru@criwi.org.cn; 2Research Institute of Wood Industry, Chinese Academy of Forestry, Haidian, Beijing 100091, China; zhw825993710@163.com; 3Hubei Technology Exchange, Wuhan 430071, Hubei, China; likang@51kehui.com

**Keywords:** wood, species, natural sunlight, artificial xenon light, aging

## Abstract

To investigate the relationship between sunlight and artificial light sources on the weathering of wood, three woods, namely, *Tectona grandis* L.F. (teak), *Stereospermum colais* (mabberley), and *Dicorynia guianensis* (basralocus), were tested under natural sunlight for 733 days and artificial xenon light for 180 h, respectively. A comparison between sunlight and artificial xenon light was made based on surface color changes at various intervals. The results showed that the woods suffered from more severe aging in the artificial xenon light exposure than that in the natural sunlight exposure. At the early stage of exposure, very good relationships were found between 70 days under natural sunlight weathering and 60 h under artificial xenon light weathering. Compared with natural sunlight, about a 30 times faster aging process was identified in the artificial xenon light. However, the linear relationship vanished at the later aging stage. It was found that the color change fluctuated in natural sunlight, while it increased steadily in artificial xenon light. The wood species affected the aging of woods. In natural sunlight exposure, the color change decreased in the order of mabberley > teak > basralocus, while in artificial xenon light exposure, color change decreased in the order of mabberley > basralocus > teak due to the easier volatilization of extractives in artificial xenon light than in natural sunlight.

## 1. Introduction

The weathering of wood limits its utilization because of color changes and visible cracks occurring on its surface. There are a number of environmental parameters that contribute to the aging of wood, such as solar radiation, moisture, oxygen, and temperature [1,2,3]. Among these factors, UV radiation (295–400 nm) is the most damaging element causing these changes at wood surfaces [4]. Although the UV spectrum represents only 5% of the energy in sunlight, its strong effect on the wood degradation process is well documented [5]. UV light can provide sufficient energy to break the major chemical bonds of wood components, which is identified as demethoxylation. The rapid surface photodegradation occurs in a few hours of exposure to accelerated weathering [6] or within a few days of natural weathering [7], and the processes are characterized by some physical and chemical changes, such as color, gloss, and wettability changes [8,9,10]. Numerous studies investigated the physical and chemical changes of wood during weathering and came to a conclusion that lignin was the most sensitive component contributing 80% to 95% of the UV absorption coefficient [11,12,13]. Absorption of light by the groups of α-carbonyl, biphenyl, and ring-conjugated double bond structures in lignin initiates the formation of free radical species and these free radicals react with oxygen to form chromophoric groups, such as carbonyl and carboxyl groups, that ultimately lead to the discoloration of wood [14].

The wood species is another key factor affecting the degradation rate of wood, such as softwood and hardwood. Hardwood and softwood differ in several aspects, like fiber dimensions, chemical component composition, lignin content, and microstructures [15,16]. Some researchers also identified that wood extractives can act as UV absorbers to protect wood from photodegradation and to reduce surface color changes [17,18,19]. A large number of comparative investigations on wood species during weathering have been carried out [20,21,22]. For example, Pandey [16] studied the surface chemistry changes of both softwood and hardwood during photo irradiation. According to the results, hardwoods underwent faster degradation than softwoods because the syringyl structure in hardwoods degraded faster than the guaiacyl structures in softwoods. 

A previous study [23] tested the color, gloss, and chemical changes of three woods at different sections under a long natural sunlight exposure. The results suggested the discoloration of the tangential section was more obvious than the radial and cross section. To save the experiment time and to quickly understand the weathering effect on the physical and chemical changes of wood, the artificial accelerated weathering condition was chosen as an alternative and showed promising results [24,25,26]. However, to completely substitute natural sunlight aging by xenon light evaluation, there should be a liner relation, while the study of the relationship between sunlight and artificial light sources on the weathering of wood was weak. Chang and Chang [17] investigated the correlation between softwood discoloration induced by accelerated light fastness testing and indoor exposure, and obtained about 250 times more severe values of lightfastness. However, the woods tested in the study were all softwood, which was structurally simple compared with hardwood. Therefore, in this study, the three woods of teak (*Tectona grandis* L.F.), mabberley (*Stereospermum colais*), and basralocus (*Dicorynia guianensis*) were chosen for their wide application in wood based flooring, and respectively exposed under natural sunlight for 733 days and artificial xenon light for 180 h. A comparison between sunlight and artificial xenon light was studied based on the surface color changes at intervals. The surface chemical changes of the woods were characterized by Fourier transform infrared (FTIR) spectroscopy equipped with an attenuated total reflection (ATR). The results were helpful to an equivalent analysis of the weathering of natural sunlight by accelerated artificial xenon light in practice.

## 2. Materials and Methods 

### 2.1. Materials

Three kinds of defect free wood samples were obtained from the market. Three replicates of wood samples were respectively prepared at tangential sections from the same board with dimensions of 45 (length) × 15 (width) × 10 (thickness) mm^3^. The parameters of the wood samples are listed in Table 1.

### 2.2. Natural Sunlight Weathering

The natural sunlight irradiation was carried out from December 11, 2013 to January 15, 2016 for a total of 733 days. The wood samples were stored in a room with a tangential section surface attached to the window and exposed under sunlight. The room temperature varied from 18 to 25 °C and relative humidity (RH) varied from 30% to 65%, where the moisture content changes of woods were about 3%. The test was interrupted after 20, 38, 70, 270, 470, 548, and 733 days of exposure and samples were taken out for testing.

### 2.3. Artificial Xenon Light Weathering

The artificial xenon light irradiation was carried out according to ISO 4892-2:2013(E) [27] with a soft irradiation source. A xenon light at 180 W/m^2^, in the range of 300 to 400 nm, at 65 °C (black panel) and 50% RH in a commercial chamber (ATLAS MTT GmbH, Hamburg, Germany) was used. The tangential sections of samples were stored about 60 cm from the lamp. The test was interrupted after 5, 15, 30, 60, 90, 120, and 180 h of exposure and samples were taken out for testing.

### 2.4. Colorimetric Analysis

The surface color of wood samples for different periods was measured by a colorimeter (CM-2600d, Konica Minolta Inc, Kyoto, Japan). According to the CIELab standard [28], the *L**, *a**, *b** color coordinates were calculated based on three independent specimens at three different positions. The color difference (Δ*E*) was calculated according to:(1)$$\Delta E=\sqrt{\Delta L{*}^{2}+\Delta a{*}^{2}+\Delta b{*}^{2}}$$
where, Δ*L**, Δ*a**, and Δ*b** were the total changes of the *L**, *a**, and *b** values during weathering testing, respectively. An increase/decrease in the *L** value means the color of the sample becomes brighter/darker. A positive Δ*a** signifies a color shift toward red, and a negative Δ*a** signifies a color shift toward green. A positive Δ*b** signifies a shift toward yellow, and a negative Δ*b** signifies a shift toward blue [29].

### 2.5. ATR-FTIR Analysis

The surface chemical changes of air-dried woods (with approximate moisture contents of 12%) before and after aging (natural sunlight of 733 days, and xenon light of 180 h) were monitored by a FT-IR spectrometer (BRUKER, Vertex 70v, Berlin, Germany) equipped with an attenuated total reflection (ATR). The surfaces were put in contact with the ZnSe crystal at a 45° angle of incidence.

## 3. Results and Discussion

### 3.1. Visual Observation

Figure 1 shows the surface morphologies of the three wood samples before and after aging. It can be seen that all wood samples underwent color changes after aging. These samples became darker and greener, which might be associated with the oxidation of lignin. More details are provided in the following discussions.

### 3.2. Natural Sunlight Weathering

Figure 2 shows color parameter changes of the three woods during sunlight exposure. In the natural sunlight weathering, the wavenumber was very large, which ranged from 290 to 5300 nm. However, the UV radiation (295–400 nm) was the most damaging element causing these changes at the wood surfaces. It appears the trends for all woods were almost similar. The data periodically waved in a very complex process during the exposure time and this may be because of the oxidation and degradation of lignin and the immigration of extractives. At the early stage, the degradation of lignin dominated the surface color change process at first, resulting in a color change of the woods. With increasing time, the discoloration of wood was balanced, which might be due to the existence of micro-cracks and the migration of extractives at the wood surface. After that, the extractives volatilized or degraded with increasing time. Therefore, the surface color change of wood was reversed. The explanation can be found in our early study [23]. Another explanation of the waving color change might be explained by the alternation of day and night. In the early weathering stage, it was winter. The sunshine time was about 8 h and the UV index was low, thus the weathering was not serious. With increasing exposure time, the sunshine time and UV index increased. Therefore, the aging became faster. However, owing to the alternation of day and night, the UV-protectable extractives migrated to the surface. The exposure of 270 days was a key point, where the discoloration of woods almost reached their maximum values with Δ*E* values of 24.66, 28.57, and 8.98 for teak, mabberley, and basralocus, respectively. After the 733 days of exposure, all woods showed decreases of Δ*L**, and increases of Δ*a** and Δ*b**, which suggests the woods became dark, red, and yellow, which is consistent with other results [30]. Considering the wood species, mabberley showed the biggest color change among the species, while basralocus was more stable (although it had bigger Δ*L** values than teak). Persze and Tolvaj [8] identified that the color change of hardwood was higher than that of softwood due to the extractives. It is known that teak and mabberley are extractive-rich woods. Therefore, the color change was significant.

### 3.3. Artificial Xenon Light Weathering

Figure 3 shows the color parameter changes of the three woods during artificial xenon light exposure. The wavenumber of xenon light was 300 to 400 nm, which can cause weathering to woods. Similar to the sunlight weathering process, all woods showed decreases of Δ*L**, and increases of Δ*a** and Δ*b** after 180 h of exposure. However, they suffered from more severe weathering in the artificial xenon light exposure. For example, after 30 h, the Δ*E* of teak, mabberley, and basralocus reached 3.90, 7.96, and 5.98, respectively, which is comparable to that of 38 days in the natural sunlight. Also, the Δ*E* values of the woods after 60 h in artificial xenon light were similar to those of 70 days in the natural sunlight, suggesting a 30 times faster weathering process in artificial xenon light. Another difference to the natural sunlight condition was the persistent increase or decrease of Δ*E* in the artificial xenon light weathering process, which might be caused by the different exposure temperature. This was consistent with Mitsui et al. [31], who obtained a persistent increase of Δ*E*. During artificial xenon light exposure, the woods were kept at 65 °C, which might easily cause the migration of extractives to the wood surface, and then volatilize them into the air. Thus, it cannot effectively protect the lignin from photodegradation. Another possible reason could be associated with the continuous aging process without disruption. Therefore, the lignin and extractives continuously degraded. At the early stage of exposure (<60 h), the values of Δ*L**, Δ*a**, and Δ*b** fluctuated slightly due to the balancing of the immigration of extractives to the wood surface and the degradation of lignin. After 60 h, these values continuously increased or decreased with increasing time. Two rapid periods of color changes were notable; one is the exposure of 60 h and the other one is 180 h. As for the wood species, the color changes decreased in the order of mabberley > basralocus > teak. Consistent with the results of natural sunlight weathering, the color change of mabberley was the largest. However, basralocus showed more accentuated color changes compared with teak, which is the opposite to that under natural sunlight, suggesting the extractives of basralocus are more sensitive in artificial xenon light. 

### 3.4. Correlation between Natural Sunlight and Artificial Xenon Light

Figure 4a shows the correlation of Δ*L** between natural sunlight and artificial xenon light in all data. A good relationship was found at the early stage of <70 days under natural sunlight weathering to 60 h under artificial xenon light weathering. The plots almost fitted very well. After 270 days of natural sunlight exposure, the Δ*L** ranged from –15 to –5, where it approximately equated to 180 h under artificial xenon light. When the natural sunlight exposure time increased, the two plots were irrelevant.

Figure 4b shows the correlation of Δ*a** between natural sunlight and artificial xenon light in all data. Consistent with Δ*L**, a good relationship was found at the early stage of <70 days under natural sunlight weathering to 60 h under artificial xenon light weathering. The Δ*a** increased sharply after 270 days of natural sunlight exposure, and the values ranged from 3 to 13, which was nearly comparable to 180 h of artificial xenon light. However, the values decreased in natural sunlight exposure after 470 days. Another good relationship was found at 548 days in natural sunlight to 120 h in artificial xenon light. Excluding this relationship, no clear relationship was found.

Figure 4c shows the correlation of Δ*b** between natural sunlight and artificial xenon light in all data. Compared with Δ*L** and Δ*a**, the Δ*b** values in natural sunlight varied remarkably with increasing time. This might be because the Δ*b** is the main color alteration affected by the degradation of lignin [32]. The lignin could be better protected in the case of extractives under natural sunlight than that exposed to artificial xenon light. Consistent with Δ*a**, the Δ*b** values in natural sunlight increased significantly after 270 days, and then decreased after 548 days. However, in the case of artificial xenon light, the Δ*b** values gradually increased. Good relationships were also observed at the very early stage of 70 days under natural sunlight weathering to 60 h under artificial xenon light weathering.

Figure 4d shows the correlation of Δ*E* between natural sunlight and artificial xenon light in all data. The two weathering processes showed totally different trends. In natural sunlight weathering, the data waved due to the fluctuated Δ*L**, Δ*a**, and Δ*b** values, while in artificial xenon light, the data increased steadily. As mentioned above, acceptable relationships were found at the early stage of exposure. Linear regression analysis of Δ*E* between the natural sunlight of 70 days and artificial xenon light of 60 h based on the average Δ*E* values was carried out.

The result of the linear regression analysis of Δ*E* between the natural sunlight of 70 days and artificial xenon light of 60 h is shown in Figure 5. The R^2^ for the two weathering processes are 0.97 and 0.93, respectively. The ratio of slopes suggests the accelerated weathering of artificial xenon light is greater than that in natural sunlight, nearly a 30 times faster in artificial xenon light than that in natural sunlight. Therefore, in practice, the artificial xenon light weathering method can be used to evaluate the long natural weathering of wood in a short time. However, 60 h of artificial xenon light weathering should be the limit value. Beyond this time, no linear relationship was found.

### 3.5. Surface Chemical Changes

The ATR-FTIR spectra of woods before and after aging are shown in Figure 6. Main changes were found at 1730, 1650, 1590, 1510, and 1230 cm^−1^. The peaks at 1590 and 1510 cm^−1^ were attributed to C=C aromatic skeletal vibration of lignin, and the peak at 1230 cm^−1^ was C–O stretching vibration in lignin and hemicelluloses. These peaks decreased significantly after aging both in natural sunlight and artificial xenon light. Peaks at 2920 cm^−1^ (alkane CH vibrations of methylene in cellulose), 1420 cm^−1^ (CH_2_ bending crystallized I and amorphous cellulose), and 1320 cm^−1^ (CH_2_ wagging in crystallized I cellulose) were characteristics for cellulose. These peaks almost remained in both the weathering processes, which indicated that weathering hardly affected the cellulose. The peak at 1030 cm^−1^ for the C-O stretching was associated with cellulose and its extractives, and the peak at 3360 cm^−1^ was associated with the free –OH groups. For different wood species, the intensity of the two peaks after weathering was different. Another difference among wood species was the multiple peaks around 2920 cm^−1^, which was characteristic for –CH_2_/–CH_3_ groups, such as the extractives of hydrophobic wax, resin, and fatty acids. The difference might be associated with the migration of extractives. For natural sunlight weathering, increased peaks at 1730 cm^−1^ were assigned to the C=O groups of hemicelluloses and 1650 cm^−1^ for the para-OH substituted aryl ketones and quinines of lignin could be found, indicating the oxidation of hemicelluloses and lignin during aging [33]. However, for artificial xenon light weathering, the two peaks decreased a lot, which might be caused by the leaching of extractives in artificial xenon light at high temperatures. It was interesting that in the mabberley group, after aging in artificial xenon light, a new peak was found at 1260 cm^−1^, which might be associated with the conjugation of the C–O of extractives with degraded aromatics, such as lignin. Therefore, mabberley showed bigger color changes compared with the other two woods.

## 4. Conclusions

All woods showed surface color changes during both natural sunlight and artificial xenon light exposure. The color changes of woods were significant and the trends were different in the two situations. In natural sunlight weathering, ageing fluctuated very complexly, while in artificial xenon light, it was increased gradually. At the early stage of exposure, very good linear relationships were found after 70 days under natural sunlight weathering to 60 h under artificial xenon light weathering, as shown by the fitted plots, suggesting a 30 times faster aging process in the artificial xenon light. After that, the linear relationship vanished. For different wood species, the degrees of discoloration were significantly different. Mabberley showed the biggest color change among all species, suggesting that mabberley was the most sensitive wood against weathering due to the leaching of UV-protectable extractives. However, the color changes of teak were larger than basralocus in natural sunlight weathering, while they were smaller in artificial xenon light, which might be caused by the leaching of extractives in artificial xenon light at high temperatures. In practice, the artificial xenon light weathering method was a good method to evaluate the long natural weathering of wood in a short time. However, it should not exceed 60 h. Besides, the development of a method to measure the sunlight degradation of wood in a short time should be explored in the future.

## Figures and Tables

**Figure 1 polymers-11-00709-f001:**
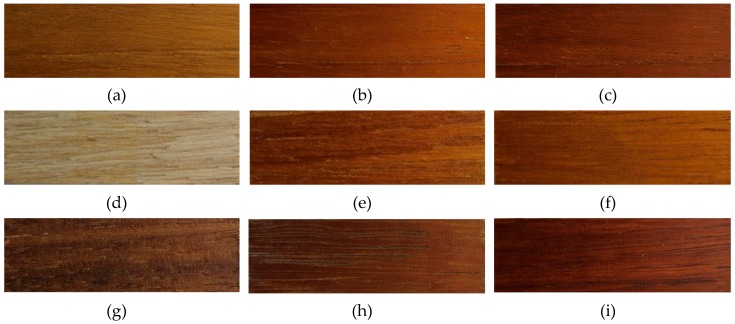
Surface morphologies of the three wood samples before and after aging. (**a**–**c**) Teak; (**d**–**f**) mabberley; (**g**–**i**) basralocus; (**a**,**d**,**g**) before aging; (**b**,**e**,**h**) after 733 days of natural sunlight aging; (**c**,**f**,**i**) after 180 h of xenon light aging.

**Figure 2 polymers-11-00709-f002:**
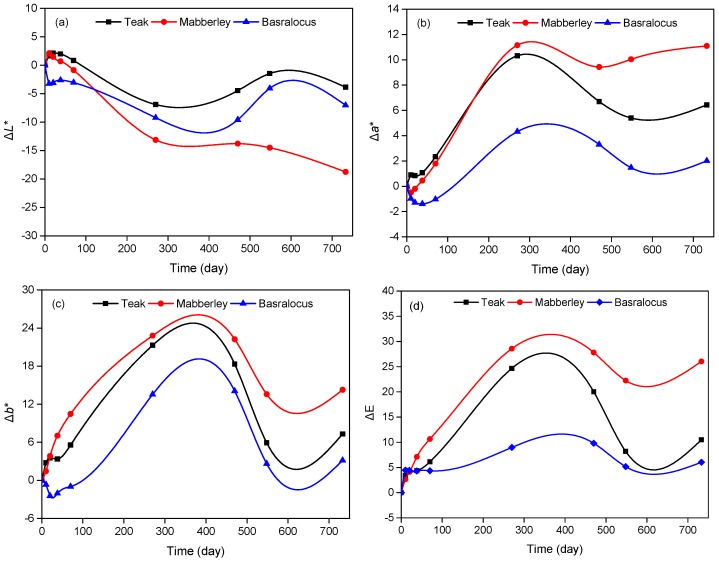
Color changes of woods during natural sunlight weathering. (**a**) Δ*L**; (**b**) Δ*a**; (**c**) Δ*b**; (**d**) Δ*E*.

**Figure 3 polymers-11-00709-f003:**
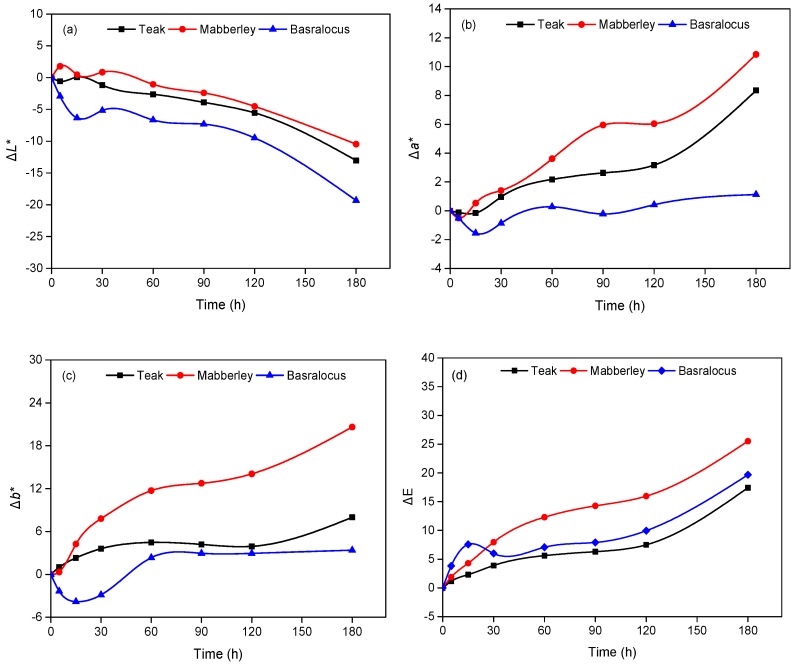
Color changes of woods during artificial xenon light weathering. (**a**) Δ*L**; (**b**) Δ*a**; (**c**) Δ*b**; (**d**) Δ*E*.

**Figure 4 polymers-11-00709-f004:**
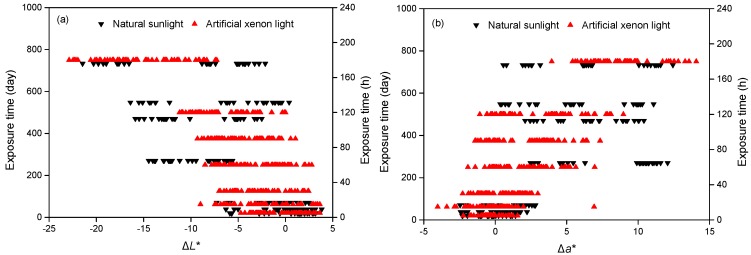
Correlation of color parameters between natural sunlight and artificial xenon light. (**a**) Δ*L**; (**b**) Δ*a**; (**c**) Δ*b**; (**d**) Δ*E*.

**Figure 5 polymers-11-00709-f005:**
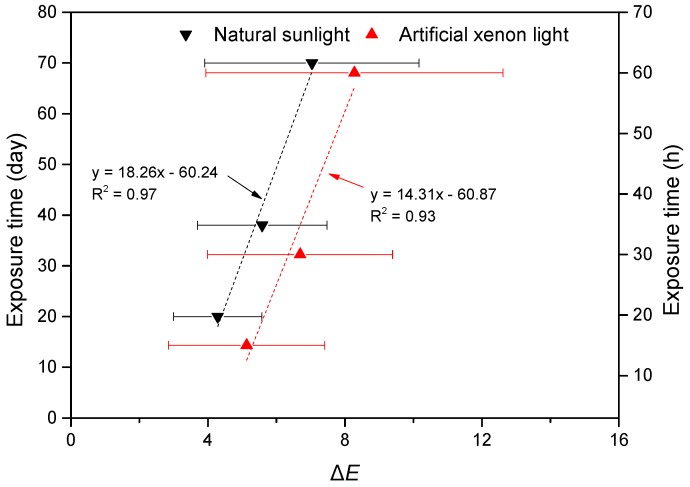
Linear regression analysis of Δ*E* between the natural sunlight of 70 days and artificial xenon light of 60 h.

**Figure 6 polymers-11-00709-f006:**
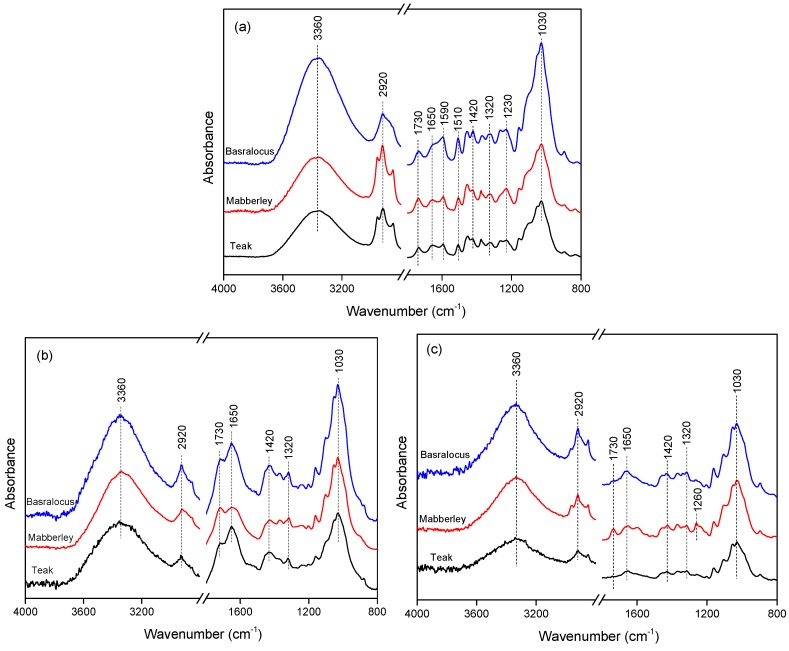
ATR-FTIR spectra of woods before and after aging. (**a**) Before aging; (**b**) after 733 days of natural sunlight aging; (**c**) after 60 h of artificial xenon light aging.

**Table 1 polymers-11-00709-t001:** Parameters of the wood samples.

Woods	Species	Density (g/cm^3^)*	Sources
Teak	*Tectona grandis* L.F.	0.62~0.68	Nature Flooring Industries, Inc, Foshan, China
Mabberley	*Stereospermum colais*	0.71~0.76	As above
Basralocus	*Dicorynia guianensis*	0.82~0.85	As above

* The values were tested at moisture contents of about 12%.

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
