# Peer review of "Comparison on the Aging of Woods Exposed to Natural Sunlight and Artificial Xenon Light"

_polymers, 2019, doi:10.3390/polym11040709_

Reviewer 1 Report

Abstract contains only experimental methods and results. I think it would be nice to include a question about why I was doing this research and an answer to that.

About method, temperature and humidity didn’t controlled. So it is hard to recognize effect of different light. 
- Comparing with xenon light, sunlight condition can be influenced by humidity and temperature
- Furthermore, sunlight condition, cross-section of sample is attached to window. If sample is contact with window directly, the condition looks like different.(In xenon light condition, 1500px gap is exist and cross-section can be exposed to atmospheres) 

3. Detailed description of existing researches seems to be insufficient in the introduction.

4. Resolution of images in Figure 1 is too low.

5. It would be nice if there were images of the wood samples showing the color change over time.

6. Need chemical approach which caused difference according wood.

7. It may be better to analyze the color change of woods depending on wood type, season, and weather (rainy or sunny).

8. There is a need to increase the content of the discussion and add conclusions.

Author Response

To Reviewer 1:

1. Abstract contains only experimental methods and results. I think it would be nice to include a question about why I was doing this research and an answer to that.

Thanks. We added the aim of this research in the abstract (line 10) that “To investigate the relationship between sunlight and artificial light sources on the weathering of wood,” Please check.

2. About method, temperature and humidity didn’t controlled. So it is hard to recognize effect of different light.

- Comparing with xenon light, sunlight condition can be influenced by humidity and temperature

Thanks. The aim of study was to make a comparison between sunlight and artificial xenon light, thus, we put woods in real environment. Although the temperature and humidity did not control, the fluctuation was small, as we mentioned in line 87, “The room temperature varied 18-25 oC and relative humidity (RH) varied 30%-65%.” In the range of temperature and humidity, the moisture content change of wood was about 3%. Thus, it caused little influence to the test. We added the moisture content change in line 88. Please check.

- Furthermore, sunlight condition, cross-section of sample is attached to window. If sample is contact with window directly, the condition looks like different.(In xenon light condition, 1500px gap is exist and cross-section can be exposed to atmospheres)

Thanks. As you mentioned, the conditions between natural sunlight and xenon light were different. Most studies considered that the xenon light was considered as a fast testing method to determine the aging of wood. However, to completely substitute the natural sunlight aging by xenon light evaluation, there should be a liner relation. Thus, to find the relation, we took this experiment but find only very good relationships were found in the early aging stage (70 days under natural sunlight weathering to 60 h under artificial xenon light). We added the aim in line 59 that “However, to completely substitute the natural sunlight aging by xenon light evaluation, there should be a liner relation, while, the study of relationship between sunlight and artificial light sources on the weathering of wood could be hardly found.” Please check.

3. Detailed description of existing researches seems to be insufficient in the introduction.

Thanks. We added details in the introduction. Please check.

Line 40: “Absorption of light by the groups of α-carbonyl, biphenyl, and ring-conjugated double bond structures in lignin initiates formation of free radical species and these free radicals react with oxygen to form chromophoric groups such as carbonyl and carboxyl groups that ultimately lead to the discoloration of wood. [14]”

Line 49: “For example, Pandey [16] studied the surface chemistry changes of both softwood and hardwood during photo irradiation. According to the results hardwoods underwent a faster degradation than softwoods because the syringyl structure in hardwoods degraded faster than guaiacyl structures in softwoods.”

Line 61: “Chang and Chang [17] investigated the correlation between softwood discoloration induced by accelerated light fastness testing and indoor exposure, and obtained about 250 times more severe in lightfastness. However, the woods tested in the study were all softwood, which was structural simple compared with hardwood.”

4. Resolution of images in Figure 1 is too low.

Thanks. We have changed the Figure 1 with high resolution. Please check.

5. It would be nice if there were images of the wood samples showing the color change over time.

Thanks. We are sorry that we did not take photo for wood samples during the aging time, but we added images for of the aged samples at the last aging stage in Figure 1. Please check.

6. Need chemical approach which caused difference according wood.

Thanks. We added ATR-FTIR results for the chemical analysis in line 241, please check. “The ATR-FTIR spectra of woods before and after aging are shown in Fig.6. Main changes were found at 1730, 1650, 1590, 1510, and 1230 cm-1. The peaks at 1590 and 1510 cm-1 were C=C aromatic skeletal vibration of lignin, and peak at 1230 cm-1 was C-O stretching vibration in lignin and hemicelluloses. These peaks decreased significantly after aging both in natural sunlight and artificial xenon light. Peaks at 2920 cm-1 (alkane CH vibrations of methylene in cellulose), 1420 cm-1 (CH2 bending crystallized I and amorphous cellulose), and 1320 cm-1 (CH2 wagging in crystallized I cellulose) were characteristics for cellulose. These peaks almost remained in both the two weathering process, which indicated the weathering affected little to cellulose. Peak at 1030 cm-1 for the C-O stretching was associated with cellulose and extractives, and peak at 3360 cm-1 was associated with the free -OH groups. For different wood species, the intensity of the two peaks after weathering was different. Another difference among wood species was the multiple peaks around 2920 cm-1, which was characteristic for –CH2/-CH3 groups such as extractives of hydrophobic wax, resin, and fatty acids. The difference might be associated with the migration of extractives. For natural sunlight weathering, increased peaks at 1730 cm-1 assigned to C=O groups of hemicelluloses and 1650 cm-1 for para-OH substituted aryl ketones and quinines of lignin could be found, indicating the oxidation of hemicelluloses and lignin during aging [33]. However, for artificial xenon light weathering, the two peaks decreased a lot, which might be caused by the leaching and unprotecting of extractives in artificial xenon light at high temperature. It was interesting that in group of Mabberley after aging in artificial xenon light, a new peak was found at 1260 cm-1 which might be associated with conjugation of the C-O of extractives with degraded aromatics, such as lignin. Therefore, Mabberley showed bigger color changes compared with other two woods.

Ref:

33. Chirkova, J.; Andersone, I.; Irbe, I.; Spince, B.; Andersons, B. Lignins as agents for bio-protection of wood. Holzforschung 2011, 65, 497-502.

7. It may be better to analyze the color change of woods depending on wood type, season, and weather (rainy or sunny).

Thanks. In this study, to eliminate other factors such as rain and sun, we put woods indoors. As for your suggestion, in the next study, we will investigate the real natural weathering to the woods, containing season, rain, sun, and so on.

8. There is a need to increase the content of the discussion and add conclusions.

Thanks. We added discussion in the manuscript. Please check.

Line 115: “Fig.1 shows the visual observation of the three wood samples before and after aging. It can be seen that all wood samples underwent color changes after aging. These samples became darker and greener, which might be associated with the oxidation of lignin. More details were provided in the following discussions.”

Line 124: “In the natural sunlight weathering, the wavenumber was very large, which ranges from 290 to 5300 nm. However, the UV radiation (295-400 nm) is the most damaging element causing these changes at wood surfaces.”

Line 128: “At early stage, the degradation of lignin dominated the surface color change process at first, resulting in color change of woods. With increasing time, the discorloration of wood was balanced, which might due to existences of micro-cracks and migration of extractives at wood surface. After that, the extractives volatilized or degraded with increasing time. Therefore, the surface color change of wood went back again. The explanation can be found in our early study [23]. Another explanation of the waving color change might be explained by the alternation of day and night. In the early weathering stage, it was winter. The sunshine time was about 8 h and the UV index was low, thus, the weathering was not serious. With increasing exposure time, the sunshine time and UV index increased. Therefore, the aging became quick. However, owing to the alternation of day and night, the UV-protectable extractives migrated to the surface.”

Line 144: “Persze and Tolvaj [8] obtained that the the color change of hardwood was higher than that of softwood due to the extractives. It was known that Teak and Mabberley were extractives-rich woods. Therefore, the color change was significant.”

Line 155: “The wavenumber of xenon light was 300-400 nm, which can cause weathering to woods.”

Line 164: “This was consistent with Mitsui et al. [31] who obtained the persistent increase of ΔE.”

Line 167: “Another possible reason could be associated with the continuous aging process without disruption. Therefore, the lignin and extractives continuously degraded.”

And we also corrected conclusion in line 268. “All woods showed surface color changes both during natural sunlight and artificial xenon light exposure. The color changes of woods were significant and the trends were different in the two situation. In natural sunlight weathering, it was in a very complex fluctuation, while in artificial xenon light, it was increased gradually. At the early stage of exposure, very good linear relationships were found of 70 days under natural sunlight weathering to 60 h under artificial xenon light weathering proved by the fitted plots, suggesting a 30 times faster aging in the artificial xenon light. After that, the linear relationship vanished. For different wood species, the degrees of discoloration were greatly different. Mabberley showed the biggest color change among all, suggesting Mabberley was the most sensitive wood against weathering due to the leaching of the UV-protectable extractives. However, the color changes of Teak was larger than Basralocus in natural sunlight weathering, while smaller in artificial xenon light, which might be caused by the leaching of extractives in artificial xenon light at high temperature. In practice, the artificial xenon light weathering method was a good method to evaluate the long natural weathering of wood in a short time. However, it should not be exceeded 60 h. Besides, the development of a method to measure the sunlight degradation of wood in short time should be highlighted.”

Ref:

8. Persze, L.; Tolvaj, L. Photodegradation of wood at elevated temperature: Colour change. J. Photochem. Photobiol. B: Biol. 2012, 108, 44-47.

23. Liu, R.; Pang, X.; Yang, Z. Measurement of three wood materials against weathering during long natural sunlight exposure. Measurement 2017, 102, 179-185.

31. Mitsui, K.; Takada, H.; Sugiyama, M.; Hasegawa, R. Changes in the properties of light-irradiated wood with heat treatment Part 1. Effect of treatment conditions on the change in color. Holzforschung 2001, 55, 601-605.

Sincerely yours,

Apr 8, 2019

Ru Liu, Hanwen Zhu, Kang Li, and Zhong Yang

Reviewer 2 Report

This manuscript coauthored by Yang and coworkers reported the correlation between natural sunlight and artificial acerbated UV photoirradiation damage to different types of commonly used woods. It was found that there is no liner correlation between two methods after longer exposure and this is very informative reference. A couple of points are recommended to be addressed:

1.    Fig 1. Needs to increase resolution. In addition, more pictures during the exposure time intervals and especially the final samples should be included.

2.    Natural sunlight and Xenon wavelength spectrum should be overlayed and included in the manuscript. A couple of typical representative ones covering all four seasons are suggested to be included. the first 70 days mostly cover winter times in this manuscript. Natural sunlight is fluctuating, and the situation is complex for sure. the authors are better to discuss a little bit more detailed instead of just telling the audience is complicated which is obviously everyone knows. Beyond this, sunshine hours and the UV index during the experimental time range are also a key reference information and it is highly recommended to be included in the manuscript.

Author Response

To Reviewer 2:

This manuscript coauthored by Yang and coworkers reported the correlation between natural sunlight and artificial acerbated UV photoirradiation damage to different types of commonly used woods. It was found that there is no liner correlation between two methods after longer exposure and this is very informative reference. A couple of points are recommended to be addressed:

1. Fig 1. Needs to increase resolution. In addition, more pictures during the exposure time intervals and especially the final samples should be included.

Thanks. We changed figure 1 with high resolution. We are sorry that we did not take photo for wood samples during the aging time, but we added images for of the aged samples at the last aging stage in Figure 1. Please check.

2. Natural sunlight and Xenon wavelength spectrum should be overlayed and included in the manuscript. A couple of typical representative ones covering all four seasons are suggested to be included. the first 70 days mostly cover winter times in this manuscript. Natural sunlight is fluctuating, and the situation is complex for sure. the authors are better to discuss a little bit more detailed instead of just telling the audience is complicated which is obviously everyone knows. Beyond this, sunshine hours and the UV index during the experimental time range are also a key reference information and it is highly recommended to be included in the manuscript.

Thanks. We added wavelength spectrum of natural sunlight and xenon into the discussion. Line 124. “In the natural sunlight weathering, the wavenumber was very large, which ranges from 290 to 5300 nm. However, the UV radiation (295-400 nm) is the most damaging element causing these changes at wood surfaces.” And Line 155The wavenumber of xenon light was 300-400 nm, which can cause weathering to woods.

We added the reason for the fluctuation of natural sunlight in line 128. “At early stage, the degradation of lignin dominated the surface color change process at first, resulting in color change of woods. With increasing time, the discorloration of wood was balanced, which might due to existences of micro-cracks and migration of extractives at wood surface. After that, the extractives volatilized or degraded with increasing time. Therefore, the surface color change of wood went back again.

Besides, as your suggestion, the sunshine hour should be an important factor to the natural weathering. We added reason for it in line 133. “Another explanation of the waving color change might be explained by the alternation of day and night. In the early weathering stage, it was winter. The sunshine time was about 8 h and the UV index was low, thus, the weathering was not serious. With increasing exposure time, the sunshine time and UV index increased. Therefore, the aging became quick. However, owing to the alternation of day and night, the UV-protectable extractives migrated to the surface.” And line 167. “Another possible reason could be associated with the continuous aging process without disruption. Therefore, the lignin and extractives continuously degraded.”

Sincerely yours,

Apr 8, 2019

Ru Liu, Hanwen Zhu, Kang Li, and Zhong Yang

Reviewer 3 Report

The work shows logical train of thought. It present a comparison method for real sun ligth degradation of wood samples compared with artificial ligth degradation which is very interesting for the scientific comunity as well as for the industrial sector. In my opinion the manuscript can be accepted after minor revisions:

The quality of the Figure 1 should be improved. 

The values reported in Table 2 and 3 should be presented in a figure n order to better show the tendency of each parameter.

The colorimetric results should be compared with the literature.

In the conclusion section the results should be better discussed in order to highligth the development of a method that allows to measure the sunligth degradation of wood in short times.

Author Response

To Reviewer 3:

The work shows logical train of thought. It present a comparison method for real sun light degradation of wood samples compared with artificial light degradation which is very interesting for the scientific community as well as for the industrial sector. In my opinion the manuscript can be accepted after minor revisions:

The quality of the Figure 1 should be improved.

Thanks. We corrected Figure 1 with high resolution. Please check.

The values reported in Table 2 and 3 should be presented in a figure in order to better show the tendency of each parameter.

Thanks. We changed the Table 2 and 3 into Figure 2 and 3. Please check.

The colorimetric results should be compared with the literature.

Thanks. We added comparison in the manuscript. Please check.

Line 144: Persze and Tolvaj [8] obtained that the the color change of hardwood was higher than that of softwood due to the extractives. It was known that Teak and Mabberley were extractives-rich woods. Therefore, the color change was significant.

Line 164: This was consistent with Mitsui et al. [31] who obtained the persistent increase of ΔE.

Ref:

8. Persze, L.; Tolvaj, L. Photodegradation of wood at elevated temperature: Colour change. J. Photochem. Photobiol. B: Biol. 2012, 108, 44-47.

31. Mitsui, K.; Takada, H.; Sugiyama, M.; Hasegawa, R. Changes in the properties of light-irradiated wood with heat treatment Part 1. Effect of treatment conditions on the change in color. Holzforschung 2001, 55, 601-605.

In the conclusion section the results should be better discussed in order to highlight the development of a method that allows to measure the sunlight degradation of wood in short times.

Thanks. We added discussion of the development of a method to measure the sunlight degradation of wood in short times. “Besides, the development of a method to measure the sunlight degradation of wood in short time should be highlighted. (line 282).

Sincerely yours,

Apr 8, 2019

Ru Liu, Hanwen Zhu, Kang Li, and Zhong Yang

Round  2

Reviewer 1 Report

The author corrected the manuscript by reflecting the eight comments I pointed out. The manuscript has been improved significantly. However, there are still some minor things to fix.

1. Please use 'Figure N' instead of 'Fig.N' in figure captions.

2. The text 'Figure 1' is missing in the caption of Figure 1.

3. Please resize the images in Figure 1 similar to each other.

4. There are two discussions in chapter 3 and 4. Chapter 4 is recommended to rewrite as the conclusion.

Author Response

We must thank you for the valuable comments and suggestions, which helped improve our manuscript greatly. Please do forward our heartfelt thanks to the reviewers. Based on the comments we received, careful modifications have been made to the manuscript. All changes were marked in red text. We hope that the revised manuscript answered the questions. Below you will find our point-by-point responses to the comments/ questions:

To Reviewer 1:

The author corrected the manuscript by reflecting the eight comments I pointed out. The manuscript has been improved significantly. However, there are still some minor things to fix.

1. Please use 'Figure N' instead of 'Fig.N' in figure captions.

Thanks. We have change “Fig.” to “Figure”. Please check.

2. The text 'Figure 1' is missing in the caption of Figure 1.

Thanks. We have corrected the caption of Figure 1 in the text. Please check.

3. Please resize the images in Figure 1 similar to each other.

Thanks. We have resized the images in Figure 1 with same sizes. Please check.

4. There are two discussions in chapter 3 and 4. Chapter 4 is recommended to rewrite as the conclusion.

Thanks. We have corrected Chapter 4 to “Conclusion”. Please check.

Sincerely yours,

Apr 16, 2019

Ru Liu, Hanwen Zhu, Kang Li, and Zhong Yang
